# Cryo-EM structure of aerolysin variants reveals a novel protein fold and the pore-formation process

Ioan Iacovache[1], Sacha De Carlo[2], Nuria Cirauqui[3,4], Matteo Dal Peraro[3,5], F. Gisou van der Goot[6] & Benoît Zuber[1]

Owing to their pathogenical role and unique ability to exist both as soluble proteins and transmembrane complexes, pore-forming toxins (PFTs) have been a focus of microbiologists and structural biologists for decades. PFTs are generally secreted as water-soluble monomers and subsequently bind the membrane of target cells. Then, they assemble into circular oligomers, which undergo conformational changes that allow membrane insertion leading to pore formation and potentially cell death. Aerolysin, produced by the human pathogen *Aeromonas hydrophila*, is the founding member of a major PFT family found throughout all kingdoms of life. We report cryo-electron microscopy structures of three conformational intermediates and of the final aerolysin pore, jointly providing insight into the conformational changes that allow pore formation. Moreover, the structures reveal a protein fold consisting of two concentric β-barrels, tightly kept together by hydrophobic interactions. This fold suggests a basis for the prion-like ultrastability of aerolysin pore and its stoichiometry.

[1] Laboratory of Experimental Morphology, Institute of Anatomy, University of Bern, Baltzerstrasse 2, 3000 Bern 9, Switzerland. [2] FEI Company, Achtseweg Noord 5, 5651 GG Eindhoven, The Netherlands. [3] Institute of Bioengineering, School of Life Sciences, Ecole Polytechnique Fédérale de Lausanne (EPFL), 1015 Lausanne, Switzerland. [4] Department of Pharmaceutical Biotechnology, Universidade Federal do Rio de Janeiro, 21941-902 Rio de Janeiro, Brazil. [5] Swiss Institute of Bioinformatics, 1015 Lausanne, Switzerland. [6] Global Health Institute, School of Life Sciences, Ecole Polytechnique Fédérale de Lausanne (EPFL), 1015 Lausanne, Switzerland. Correspondence and requests for materials should be addressed to F.G.v.d.G (email: gisou.vandergoot@epfl.ch) or to B.Z. (email: benoit.zuber@ana.unibe.ch).

Aerolysin is a major contributor to the pathogenicity of *Aeromonas* spp, which causes gastroenteritis, deep wound infections and sepsis in humans[1]. It is a pore-forming toxin (PFT), which is secreted by the bacterium as a water-soluble protein, binds receptors on their target cell membrane and, following proteolytic activation, forms circular heptameric oligomers[2] that insert into the plasma membrane thus permeabilizing it, potentially leading to cell death. Aerolysin (Supplementary Fig. 1a) belongs to the family of β-PFTs, meaning that the final pore spans the membrane in the form of a β-barrel[2] and defines the aerolysin-like family of proteins that share a common structural motif[3] (Supplementary Fig. 1b). The structure of the secreted soluble 52-kDa monomer was solved by X-ray crystallography two decades ago and shows that the protein is composed of four domains[2] (Supplementary Fig. 1a). Domains 1 and 2 are responsible for the dual binding to *N*-glycosylated glycosylphosphatidylinositol (GPI)-anchored proteins, which act as aerolysin receptors[4] with domain 2 binding directly to the glycan core of the GPI-anchor, while domain 1 is responsible for binding the *N*-linked sugar modifications present on the receptor. Domain 3 consists of a five-stranded β-sheet and a prestem loop, which is curled up against the β-sheet. Previous experiments have shown that the prestem loop ultimately refolds into one hairpin of the final transmembrane β-barrel[5], and is responsible for driving both the insertion and the anchoring of the β-barrel. Domain 4 is a prolongation of the domain 3 β-sheet, but the sheet is split open by the C-terminal peptide (CTP) into a twisted double β-sheet fold (Fig. 1f; Supplementary Fig. 1)[2,6]. The CTP is a propeptide present in proaerolysin, the form secreted by the bacterium. It must be subsequently cleaved by host or bacterial proteases for the toxin to be active. While following CTP removal, the wild-type toxin spontaneously oligomerizes, and transitions to the membrane-inserted pore state[7], various mutations have been identified that arrest pore formation at different stages along the pathway. The structures of these mutants were coined prepore, post prepore and quasipore[8,9]. Low-resolution cryo-EM analysis (16.6–18.3 Å) coupled to dynamic integrative modelling[10] provided pseudo-atomic models of these intermediate structures, indicating profound rearrangements of domains 3 and 4 during pore formation[9]. Benefiting from the recently developed direct electron detector technology[11,12], we now provide near-atomic resolution cryo-electron microscopy (cryo-EM) maps (3.9–4.5 Å) of the aerolysin prepore, post-prepore and quasipore states. We also obtained a 7.9-Å resolution structure of the final, wild type, aerolysin pore in detergent. Combined, these structures, consistent with the recent lysenin pore structure[13], describe the major steps involved in the pore-formation process by aerolysin: the transition from the monomer to the oligomer with the formation of two highly stable concentric β-barrels, the zipper-like formation of the β-barrel and the final piston-like puncturing of the lipid bilayer. These results most likely apply to the entire aerolysin family. Furthermore, the novel concentric β-barrel fold supports a hypothetical model of transmembrane pores formed by prion-like proteins[14,15].

## Results

**Structure of the aerolysin prepore.** While the wild-type aerolysin pore is a membrane-inserted heptamer, a single-point mutant Y221G was found to arrest the protein in its prepore stage. The cryo-EM map of Y221G (Fig. 1a,b; Supplementary Fig. 2a), shown previously to be a head-to-head dimer of heptamers in solution[8,9] was here determined at a global resolution of 3.9 Å, extending to 3.0–3.5 Å in most parts of the complex (Supplementary Fig. 3a,c,d), allowing most bulky side chains to be resolved (Supplementary Fig. 4a). We could thus derive an accurate atomic model of the aerolysin prepore (Fig. 1c,d).

Comparison of the structure of soluble aerolysin[2,9] (Supplementary Fig. 1a) with that of one protomer in the prepore reveals the reorganization involved in the oligomerization (Fig. 1e; Supplementary Movie 1). Domain 1, which is connected to domain 2 by a long amino-acid stretch, rotates by 180° and moves by 30 Å. Following this movement, all receptor-binding sites become ideally located with respect to the target membrane (D1, D2 and dashed line in Fig. 1c,d). In particular, half of domain 1, which protrudes from the tight circular oligomer, bears the receptor-binding site (D1 in Fig. 1c,d) and contains a few less well-resolved loops, which are likely highly mobile (Leu11-Lys22, Leu50-Trp54 and Gly68-Asn72; Supplementary Fig. 3c,d; Supplementary Fig. 4b). The non-protruding part of domain 1 forms contacts with domain 2 from the same protomer, as well as with domain 2 of the adjacent protomer. This interaction in particular involves His132 and Glu64 (purple in Fig. 1c,d, and ball and stick in Supplementary Fig. 4c), explaining why the protonation of His132 is required for oligomerization to occur[6,16].

Domains 2 and 3 are mostly unchanged with respect to the soluble structure (1.3 and 3.4 Å root mean squared deviations, respectively). In particular, the prestem loop remains locked in the same curled-up position (yellow loop in Fig. 1e).

A previously unknown and most interesting change occurs in domain 4. As expected, the CTP is not part of the heptamer, its release being the limiting step in the pore-formation process[7]. Removal of the CTP uncovers a hydrophobic patch at the extremity of a five-stranded β-sheet, which had initially led to consider domain 4 as the membrane-spanning domain[2]. Upon CTP removal, the β-strands rearrange to form a β-sandwich in the core of which the hydrophobic residues are buried (Fig. 2; Supplementary Fig. 5a,b). Furthermore, domain 4 straightens with respect to domain 3 and moves by 20 Å (turquoise domain in Fig. 1e). This movement allows H-bonding between β-sandwiches from two monomers, resulting in aerolysin oligomerization (box in Fig. 1c; D4 box in Supplementary Fig. 6c). This circular association of β-sandwiches leads to the formation of a novel protein fold consisting of two concentric β-barrels held together by hydrophobic interactions (box in Figs 1c and 2a,b,c). The inner barrel (tilt angle: 35.5°, shear number: 14 and radius: 13.7 Å) is formed of 14 anti-parallel β-strands, consistent with the fact that each β-hairpin originates from the prestem loop[5,17]. Interestingly, the outer barrel is a 21-stranded β-barrel (tilt angle: 43.7°, shear number: 28 and radius: 23.1 Å), each protomer contributing three strands. The first two strands show the typical anti-parallel fold found in the majority of β-barrels, as well as in the inner barrel. The third β-strand, formed by C-terminal residues 410–416, forms an anti-parallel contact with the second β-strand of the same protomer and a parallel contact with the first β-strand of the next protomer. To our knowledge, the only reported β-barrels formed by an odd number of strands are the members of the voltage-dependent anion channel family[18]. The high-resolution aerolysin prepore structure, thus reveals an elegant novel fold consisting of two concentric β-barrels held together by interaction of hydrophobic side chains. This hydrophobically glued double barrel provides the explanation for the extraordinary stability of the aerolysin oligomer, which resists days of exposure to high concentrations of chaotropic agents, boiling in SDS and even high-temperature proteolysis[19]. This extreme stability was previously attributed to the formation of the transmembrane β-barrel. We now show that both mutants used in this study are as resistant to proteolysis and high temperatures as the wild-type protein (Supplementary Fig. 5c) even though the transmembrane β-barrel does not form.

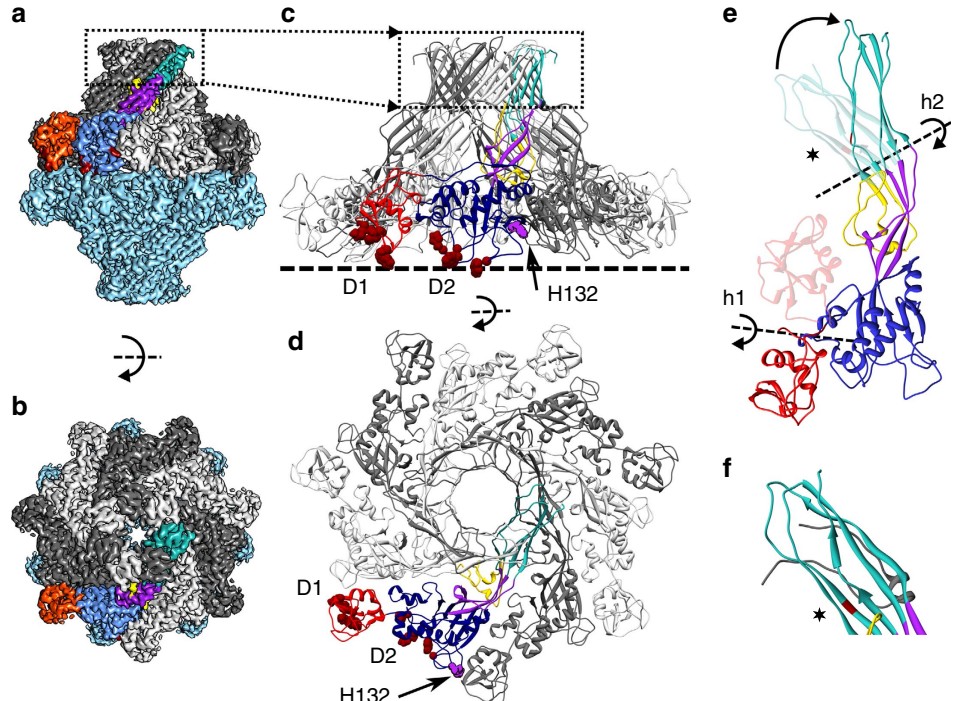

**Figure 1 | Prepore architecture and initial rearrangements required for oligomerization.** (**a**) Side view of the aerolysin prepore cryo-EM map. The map shows two prepore heptamers stacked head-to-head. One of the heptamers is shown in light blue, while the second heptamer is shown with alternating grey levels for succeeding monomers. One subunit is colour coded with the densities corresponding to domain 1 in red, domain 2 in blue, domain 3 in purple and domain 4 in turquoise. The positions of the receptor-binding sites are highlighted in dark red. (**b**) Top view of the same cryo-EM map as in **a**. Dashed rectangle marks the position of the concentric β-barrel fold. (**c**) Side view of the prepore heptamer structure with alternating subunits coloured in different grey levels. One subunit is coloured as in **a**, highlighting domains and residues of interest. Receptor-binding sites in domains 1 and 2 are labelled D1 and D2, respectively. Of note is the position of H132 (in purple) as an inter-subunit contact (see text). Dashed line shows the position of the membrane. (**d**) Heptamer top view. (**e**) One subunit of the prepore coloured as in **a** superposed over the previously published soluble aerolysin Y221G monomer X-ray structure[9] (PDB code: 3C0N—shown semi-transparent) using domain 2 as anchor. For clarity, domains 2 and 3 are shown for the prepore only. The transition from the soluble structure to the prepore requires a 180° rotation of domain 1 around hinge 1 (h1), as well as a straightening and partial rearrangement of domain 4 around hinge 2 (h2) upon removal of the CTP (not shown). The position of the Y221G mutation is shown in dark red and indicated by a star. (**f**) Detail of the soluble monomer showing domain 4 as in **e**, as well as the CTP in grey.

Interestingly, the concentric barrel fold also provides an explanation as to why aerolysin oligomers are heptameric. We have analysed the various stoichiometries that would allow the formation of a double barrel with reasonable geometry by modulating the tilt angles of the β-strands in the inner and outer barrels (Supplementary Fig. 5d), and found that the structure with the minimal number of protomers that fulfils this requirement is a heptamer. Closer analysis of the recently solved cryo-EM structure of lysenin[13], an aerolysin family member, indicates that its nonameric pore also has a concentric β-barrel arrangement, though 25% shorter (Fig. 2d,e), showing as predicted (Supplementary Fig. 5d) that higher stoichiometries are also possible, and are likely determined by the conformation and nature of additional flanking domains of aerolysin-like PFTs. Like aerolysin, lysenin oligomers are resistant to dissociation in SDS, although only at room temperature[20]. Interestingly, aerolysin concentric β-barrels, which are characterized by a very hydrophobic core (Fig. 2c; Supplementary Fig. 5a), are reminiscent of a hypothetical model proposed for amyloid pores[14,15]. Aerolysin structure could therefore contribute to a better understanding of the conformations adopted by amyloid peptides in solution and in a membrane environment.

**Structure of aerolysin post prepore**. We next analysed the structure of a second aerolysin mutant (K246C/E258C; Supplementary Fig. 2a; Supplementary Fig. 7). While the Y221G mutation, which is positioned just two amino acids downstream of the inner β-barrel (star in Fig. 1e), prevents the prestem loop from moving away from the five-stranded sheet in domain 3, the second mutant leads to a block at later stages, namely, hindering the formation of the full transmembrane β-barrel. This is due to the presence of an engineered disulphide bridge within the prestem loop between the residues 246 and 258 (ref. 5).

As we have previously described, this mutant forms a head-to-head dimer of heptamers in solution, in which one heptamer is in a prepore conformation somewhat beyond that seen for the Y221G mutant (referred to as post prepore) and the second heptamer has almost reached completion of the pore (referred to as quasipore)[9]. Most likely, the introduced cysteine bridge does not only prevent the full extension of the pore β-barrel, but we speculate that it also slows down the transition from prepore to post prepore and quasipore. Thus, at high concentrations ($\sim 1\,mg\,ml^{-1}$), the protein heptamerizes into prepores that dimerize, as Y221G heptamers. This is followed by the progression to the post-prepore and quasipore states. Due probably to stochastic difference in kinetics and steric constraints, only one of the heptamers is able to undergo quasipore formation with extension of the inner β-barrel, leaving just enough space for the second heptamer in the dimer to undergo progression to the post-prepore state (Supplementary Fig. 7f). The resolution of the obtained EM map is 4.5 Å and locally extends from 3.5 to 4.5 Å (Supplementary Fig. 3a,e,f).

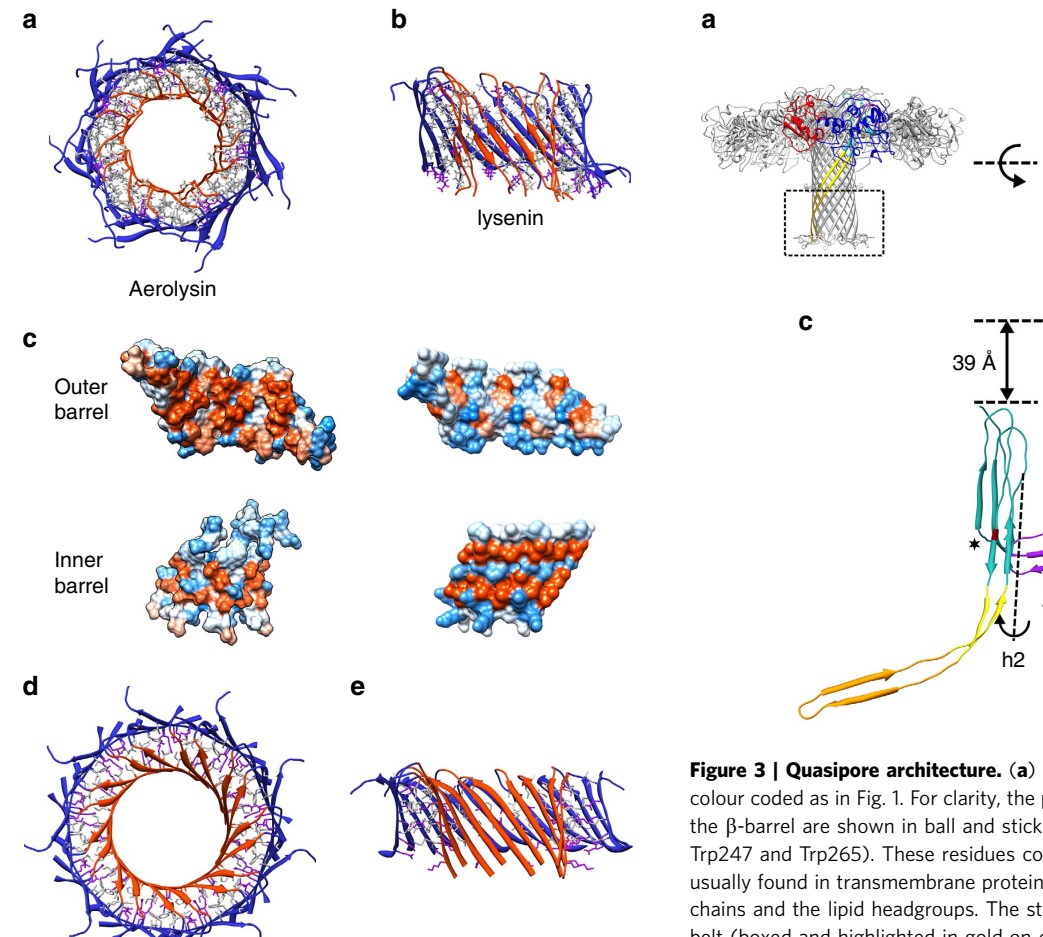

**Figure 2 | Concentric β-barrel fold.** (**a**) Top view of the concentric β-barrels (dashed rectangle in Fig. 1a,c), highlighting the extensive hydrophobic network between the two barrels. The outer and the inner β-barrel backbone are drawn in blue and orange, respectively. Hydrophobic residues are shown in dark grey, uncharged/indifferent residues are shown in light grey and charged residues are shown in purple. Note that only the side chains pointing in the space between the barrels are drawn. (**b**) Side view of the concentric β-barrel fold as in **a**. The three front subunits are not displayed for clarity. (**c**) Hydrophobicity according to Kyte *et al.* (kdHydrophobicity)[43] of the outer barrel inner surface (top left) and outer surface (top right), as well as of the inner barrel inner surface (bottom left) and outer surface (bottom right). Colours range from blue (most hydrophilic, kdHydrophobicity: − 4.5) to white (kdHydrophobicity: 0) and to orange red (most hydrophobic, kdHydrophobicity: 4.5). (**d**,**e**) Top and side views of the lysenin concentric β-barrel fold as in **a** and **b**.

The analysis of the post-prepore structure (Supplementary Fig. 8) indicates that upon progression towards pore formation, domains 3 and 4 twist slightly with respect to domains 1 and 2 (root mean squared deviations of 4.2 and 4.8 Å, respectively), allowing elongation of the internal β-barrel by incorporation of more amino acids, including tyrosine 221 (star in Supplementary Fig. 8c). The density corresponding to the prestem loop partially disappears from the pocket between the subunits, as it becomes unstructured and flexible. The flexibility of the prestem loop in this conformation prevents accurate modelling, though a clear low-resolution density between the β-barrel and domain 3 suggests that it is still loosely interacting with domain 3 (gold density in Supplementary Fig. 7d,e).

**Figure 3 | Quasipore architecture.** (**a**) Quasipore heptamer side view colour coded as in Fig. 1. For clarity, the position of the aromatic residues in the β-barrel are shown in ball and stick representation (Tyr233, Phe245, Trp247 and Trp265). These residues constitute the aromatic belt that is usually found in transmembrane proteins at the interface between the acyl chains and the lipid headgroups. The structure beyond the first aromatic belt (boxed and highlighted in gold on one monomer) is not folded in the quasipore structure and was modelled by homology to the anthrax protective antigen (see text). (**b**) Quasipore heptamer top view. (**c**) Quasipore subunit superposed over the post-prepore subunit (semi-transparent) using domain 2, as anchor with the same colour coding as in Fig. 1. For clarity, domains 1 and 2 of the post prepore are omitted. The main movements are highlighted, h2—hinge 2 and h3—hinge 3. The position of the Y221 residue is shown in dark red and marked by a star. As in **a**, the 28 residues shown in gold are not folded in the quasipore map.

**Structure of aerolysin quasipore and pore.** The transition from the post-prepore to the quasipore state involves two major conformational rearrangements of the protein. First, the inner β-barrel, which will eventually form the transmembrane pore, elongates by incorporating 15 more residues from the prestem loop (Fig. 3a,c shown in yellow). Second, the protein undergoes a vertical collapse, which translates the extended inner β-barrel and the outer β-barrel 39 Å towards the target membrane. This piston-like movement of the concentric β-barrels is enabled by the torsion of two hinge regions flanking domain 3, at the base of the outer β-barrel, which molecular dynamics previously predicted to be highly flexible (h2 and h3 in Fig. 3c)[9]. The hinge region h2 at the interface of domains 2 and 3 bends, twisting the three-stranded β-sheet belonging to the outer β-barrel. Simultaneously, a second hinge h3 at the interface of domains 3 and 4 twists to accommodate the movement, while keeping the concentric β-barrel fold intact and resulting in the translation of the concentric β-barrels by 39 Å towards the membrane (Fig. 3c). This large protein collapse is necessary for the fully extended inner β-barrel to pierce the host membrane (Figs 4 and 5; Supplementary Movie 2), which is reminiscent of the situation of lysenin[13,21] and cholesterol-dependent cytolysins[22]. The aerolysin

quasipore is a more compact structure than the prepore and post prepore: the pocket previously occupied by the prestem loop becomes an inter-subunit contact (Supplementary Fig. 6e).

To verify that the quasipore conformation is indeed representative of the full pore conformation, we analysed the structure of wild-type aerolysin heptamers. Detergent added during proteolytic activation reduced, but could not completely prevent aggregation of the fully formed pores (Supplementary Fig. 2b). If performed at sufficiently low aerolysin concentration, it allowed us to reduce aggregation and to obtain relatively well-dispersed individual pores in detergent micelles. Cryo-EM analysis led to a 7.9 Å resolution map of the wild-type pore (Fig. 4a; Supplementary Fig. 10a). This resolution was sufficient to fit the quasipore model by rigid-body docking, demonstrating that it very closely corresponds the wild-type final pore density map (Fig. 4). The 28 amino acids of the fully extended inner

β-barrel that were not resolved in the quasipore structure were modelled, using the recently solved anthrax protective antigen transmembrane barrel as structural template (gold in Figs 3 and 4; boxed in Fig. 3a; Supplementary Fig. 9). This modelled region, which is flanked by the detergent micelle density, fits also the wild-type pore density (Fig. 4). Most interestingly, the nine amino acids at the tip of each β-hairpins adopt a different conformation than in anthrax protective antigen and fold sideways in a rivet-like fashion, confirming our previous experimental conclusions[5]. Similarly, to other transmembrane β-barrels, including the lysenin pore[13], the position of two rings of aromatic residues spaced by 32 Å delineate the edge of the transmembrane regions (highlighted in Fig. 4a). The extended inner β-barrel of the pore is 87 Å long, spanning the entire length of the protein (Supplementary Fig. 4). Interestingly, a comparison of the concentric β-barrel structures of aerolysin and lysenin with other membrane-spanning β-barrels reveals a strand inversion in the β-hairpins (Supplementary Fig. 9a). While in both aerolysin family members, the N-terminal strand is on the right side of the β-hairpin (when viewed from outside of the β-barrel with the extracellular side upwards), it is on left side in all other available β-PFT structures. The reason or consequences of this strand inversion are intriguing and remain to be determined.

## Discussion

Our near-atomic resolution analysis of aerolysin at different stages towards pore formation reveals that aerolysin secreted by the bacterium is folded as a loaded spring, blocked by two pegs, the CTP and the prestem loop (Fig. 5; Supplementary Movie 1). Upon removal of the CTP, the protein straightens its fourth domain and oligomerizes, thereby generating a novel structure formed of two concentric β-barrels held together mainly through hydrophobic interactions. A second peg, the prestem loop, then gradually folds in a zipper-like manner through H-bonding and extends the inner β-barrel to an estimated length of 87 Å in the final pore conformation (Supplementary Movie 2), similar in the length to the barrel formed by anthrax protective antigen[23] and lysenin[13] (Supplementary Fig. 9b). Furthermore, the protein collapses vertically, bringing both concentric β-barrels towards the target membrane in a piston-like movement. A similar collapse also occurs upon pore formation by the unrelated family of cholesterol-dependent cytolysins[22,24], but not for anthrax

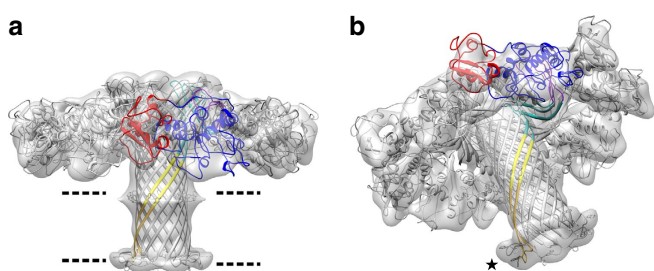

**Figure 4 | Wild-type aerolysin pore structure.** (**a**) The pore structure obtained from the quasipore map was rigid-body docked into the wild-type aerolysin map. The fit confirms that the quasipore structure indeed represents the structure adopted by the wild-type aerolysin in the pore conformation. For clarity, one monomer of the pore structure has been colour coded as in Fig. 3 with domain 1 in red, domain 2 in blue, domain 3 in purple and domain 4 in turquoise. The part of the prestem loop that was resolved in the quasipore map is shown in yellow, while the residues missing from the quasipore map are shown in gold (see text and Fig. 3). The position of aromatic residues in the β-barrel (shown in ball and stick representation) hint at the position of the two leaflets of the membrane (dashed lines). (**b**) Tilted view of **a** showing the fit of the loops at the end of the transmembrane β-barrel (asterisk). As previously reported, upon crossing the bilayer the tips of the β-barrel loops fold back forming a rivet[5].

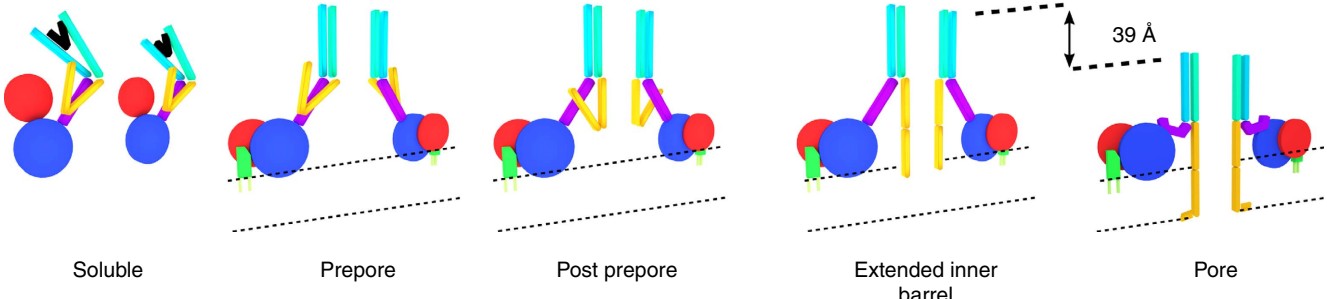

**Figure 5 | Aerolysin mode of action.** Schematics of the structural changes observed during aerolysin mode of action from the soluble inactive monomer to membrane-inserted oligomer subunit (Supplementary Movie 1). In soluble aerolysin, the CTP (black) acts as a peg blocking the protein in its inactive conformation. Removal of the CTP and oligomerization lead initially to the rotation of domain 1 (red), which together with domain 2 is now able to bind the receptor (green), as well as to a reorganization of the domain 4 into a β-sandwich, which through oligomerization becomes a concentric β-barrel (turquoise; prepore). The reorganization of domain 4 triggers the gradual extraction of the prestem loop (yellow), as more residues are folded in the stem domain (post prepore). The elongation of the stem β-barrel, as visualized in our cryo-EM structures proceeds from the top in a zipper-like manner as previously hypothesized[23]. We speculate that before the oligomer collapses, the complete prestem loop has refolded and formed an elongated 100-Å inner β-barrel, which is long enough to partially insert its hydrophobic tip into the membrane (extended inner barrel state). Subsequently, torsions at a first hinge located at the interface of domains 2 (blue) and 3 (purple), and at a second hinge at the interface of domains 3 and 4 (turquoise) lead to the collapse of the structure by 39 Å and the insertion of the inner β-barrel through the membrane, resulting in the formation of a pore, while the tips of the β-barrel loops fold back to anchor the pore as a rivet (pore).

protective antigen nor for *Staphylococcus aureus* α-hemolysin family members[23,25]. The collapse results from protein bending around two hinges located within domain 3, and at the interface of domains 3 and 4, respectively, and that we previously predicted by molecular dynamics to be highly flexible (Supplementary Fig. 10b; Supplementary Note 1)[9]. Since the prestem loop already starts to refold in the post-prepare conformation, extending the inner β-barrel while the complex has not begun to collapse, it appears that these two major conformational rearrangements occur in two successive steps. First, the inner barrel gradually and fully extends in a zipper-like manner from the top of the concentric β-barrel by incorporating the whole prestem loop. Second, the protein collapses towards the target membrane by 39 Å, thereby injecting the hydrophobic tip of the inner β-barrel into and through the membrane, leading to the pore formation[5] (Fig. 5; Supplementary Movie 2). Once in the membrane, each loop formed by nine amino acids at the tip of the inner β-barrel folds sideways and lies approximately parallel to the membrane plane, leading to a rivet-like anchoring. This confirms our predictions based on biochemical experiments[5]. All steps taking place after CTP removal and aerolysin heptamerization happen spontaneously and irreversibly[7].

Since the common motif shared by all aerolysin family members spans the third and fourth domains of aerolysin, the novel concentric β-barrel fold, as well as the here described mode of action are most likely shared by all aerolysin-like proteins[3]. Indeed, the recently solved structure of lysenin shows a similar arrangement of the β-strands[13]. Interestingly, aerolysin collapses by 39 Å, which is significantly larger than the 20 Å collapse proposed by Bokori-Brown et al.[13]. This could suggest that aerolysin binds its receptors (GPI-anchor proteins) at a distance from the target membrane that is larger than the distance between the membrane and the lysenin-binding site on its receptors (sphingomyelin), thus requiring a larger collapse for the inner β-barrel to pierce the target membrane. On the other hand, the extent of lysenin collapse was estimated based on a hypothetical lysenin prepore conformation, modelled by concatenating the X-ray structure of nine soluble monomers. Thus, the transition from lysenin prepore to pore could potentially be larger than predicted, as indicated by the ~30-Å collapse determined by atomic force microscopy[21].

The double concentric β-barrel fold that we report in aerolysin is also present, though previously not highlighted and 25% shorter, in lysenin pore[13]. Interestingly, aerolysin and lysenin concentric β-barrels are not held together by the same kind of the interactions. Whereas the space between the two aerolysin β-barrels (average distance between opposite strands backbone: 8.75 Å) is occupied to 87% by hydrophobic or uncharged side chains, only 67% of the residues found between lysenin β-barrels (average distance: 10.95 Å) are uncharged or hydrophobic (Fig. 2). This difference, together with the different stoichiometry, might explain why aerolysin can resist boiling in SDS whereas lysenin, to our knowledge, can stand SDS treatment at room but not at boiling temperature[19,20]. It has been speculated that this extreme stability also protects aerolysin from cell clearance and could explain why eukaryotic cells generally do not recover from an intoxication with aerolysin in contrast to other β-PFTs[19]. The concentric β-barrel fold might not be restricted to aerolysin family members. It is in fact reminiscent of the hypothetical model of certain amyloid transmembrane pores, which have been proposed to be formed by two concentric β-barrels maintained together exclusively by hydrophobic interactions[14,15].

In conclusion, the structures of aerolysin heptamers at four different stages of the pore-formation process provide an unprecedented understanding of the conformational changes that allow a protein to morph from a soluble into a transmembrane state. The structures also rewardingly reveal a truly novel concentric β-barrel fold that drives the stoichiometry of the complex and provides it with an extreme stability, reminiscent of amyloid oligomers, which could in turn share this fold.

## Methods

**Sample preparation and image acquisition.** Protein purification and cryo-EM sample preparation were done as previously described in the case of Y221G and K246C/E258C mutants[6]. Wild-type aerolysin was prepared similarly except that proteolysis was performed on aerolysin at a concentration of 0.4 mg ml$^{-1}$ and in the presence of 0.02% lauryl maltose neopentyl glycol (Anatrace) followed by a 5 min centrifugation at 1,000$g$ to remove aggregates. Image acquisition was done on an FEI Titan Krios (Eindhoven, the Netherlands) operated at 300 kV at the NeCEN facilities (www.NeCEN.nl), and on an FEI Tecnai F20, with an FEI Falcon 2 direct detector, at a magnification of 59,000 (pixel size: 1.34 Å) or 62,000 (pixel size: 1.68 Å) at set defoci ranging from −1.6 to −3 μm. Images were acquired automatically using EPU software (FEI) in movie mode[12]. The first frame (1.49 e$^-$/Å$^2$) was discarded, and seven frames (2.99 e$^-$/Å$^2$ each) were saved (total dose: 22.42 e$^-$/Å$^2$).

**Image processing.** All processing was performed in Relion[26]. Image movies were first averaged. Contrast transfer function (CTF) was estimated using CTFFIND3 (ref. 27) or CTFFIND4 (ref. 28). Between 1,000 and 2,000 particles were manually picked from a sub-set of the micrographs and the Relion auto-picking procedure[29] was used to pick up 56,966 particles for the Y221G mutant, 260,958 particles for the K246C/E258C mutant and 29,079 particles for the wild-type protein. The particles were subjected to two-dimensional and three-dimensional (3D) classification (using as initial models the published low-resolution maps of the same proteins[9], which were low-pass filtered at 50 Å). The particles composing the best classes (42,962 for the Y221G mutant, 70,766 for K246C/E258C mutant and 27,108 for the wild-type protein) were used for 3D auto-refinement. In the case of the wild-type protein, focused refinements were used using a mask generated from the quasipore structure. D7 symmetry was imposed for the Y221G mutant; and C7 symmetry was used for the K246C/E258C mutant and the wild-type protein. Relion particle polishing procedure for motion correction was used to generate 'shiny' particles, which were used for a final 3D auto-refinement run to generate the EM map. Resolution estimates were based on gold standard Fourier shell correlation (FSC; 0.143 cutoff) and sharpened by applying a negative B-factor using automated procedures[30]. Local resolutions were calculated using Bsoft[31,32].

**Model building.** Atomic model building into the density maps was carried out sequentially using Rosetta procedures described[33,34]. In a first step, the X-ray crystallography-derived model of aerolysin mutant Y221G soluble monomer[9] (PDB code: 3C0N) was edited in UCSF chimera[35] to obtain a PDB file containing the B chain lacking the CTP (residues 441–468 were removed). The density approximately corresponding to one aerolysin subunit was extracted from the Y221G mutant cryo-EM map by segmentation in UCSF chimera and the edited PDB model was rigid-body fitted into it using a selection of residues spanning domains 2 and 3. The model fit to density was refined in torsional space with Rosetta. The resulting model was rigid-body fitted in the original cryo-EM map and was refined in Cartesian space, while imposing D7 symmetry. To model the post-prepare structure, the whole quasipore was masked out of the K246C/E258C mutant cryo-EM map. Since the prestem loop is not resolved in the post-prepare map, we used an edited version of the prepore model where we deleted the prestem loop as an initial model for refinement in the post-prepare cryo-EM map. The post-prepare structure was similarly used as an initial model for the quasipore map. Given the poorer resolution of loops L14-N27 and W54-Y76 in domain 1, and G129-Y135, T154-N168 and L322-P337 in domain 2, these were subsequently rebuilt from the soluble aerolysin X-ray structure and the models were subjected to a final refinement in Cartesian space against the cryo-EM maps. Subsequently, we computed a feature-enhanced map using a recently introduced procedure in Phenix[36]. This procedure was developed to increase interpretability of X-ray crystallography maps and has also been shown to be useful for cryo-EM maps[37]. Note that we did not perform any model refinement based on the feature-enhanced maps. Only the map shown in Supplementary Fig. 4c is a feature-enhanced map, all the other displayed maps were B-factor-sharpened maps obtained as described in 'image processing' section above.

**Model quality assessment.** For model validation, all atoms were moved by 0.1 Å in random direction and each modified model was refined against one of the half maps obtained in Relion, after it had been B-factor-sharpened (training map)[38]. These refined atomic models were used to generate two simulated density maps using EMAN2 e2pdb2mrc.py program[39]: one for the Y221G mutant and one for the K246C/E258C mutant. For each simulated map, two FSCs were computed: a training FSC between the simulated map and the training map, and a test FSC between the simulated map and the second half map (Supplementary Fig. 5a,b). The high similarity between the training and test FSCs suggests that the models were not over-refined.

Model quality was evaluated with MolProbity[40]. The prepore, the post prepore and the quasipore obtained MolProbity scores of 1.93 (100th percentile within 3.25–4.19-Å resolution range), 2.29 (99th percentile within 3.25–4.75-Å resolution range) and 2.13 (100th percentile within 3.25–4.75 Å resolution range), respectively. These excellent scores support the validity of the models.

**Full pore model building.** The missing 28 amino acids (L235-N262) in the quasipore structure were modelled in Modeller 9.16 (ref. 41). They were created using as a template the anthrax protective antigen structure (PDB code: 3J9C). To create the new right-to-left strand disposition, the connecting loop (W247-V250) was cut and a new one was rebuilt joining the chains from right to left. This model was fitted by rigid-body docking into the wild-type aerolysin density map using UCSF chimera. Finally, the rivet loop, which was not present in the anthrax protective antigen structure, was built with Modeller 9.16 to fit the wild-type pore density map.

**Limited proteolysis.** Stability assay was performed as previously described[19]. In brief, wild-type or mutant aerolysin at 0.4 mg ml$^{-1}$ in 40 mM Tris pH 8 150 mM NaCl was activated with trypsin agarose for 1 h at 22 °C. Removal of the trypsin agarose was done by centrifugation at 1,000$g$ for 5 min and the sample was dialysed for 18 h against 30 mM HEPES pH 7.5 30 mM NaCl. The sample was heated for 10 min to 70 °C followed by the addition of thermolysin at a aerolysin:thermolysin ratio of 1:200. Proteolysis was allowed to proceed for 10 min and was followed by analysis by SDS–polyacrylamide gel electrophoresis. All reagents were acquired from Sigma-Aldrich.

**Stoichiometry assessment.** The relation between chain tilt angle in the inner β-barrel $\beta_i$, in the outer β-barrels $\beta_o$ and the number of protomer in an aerolysin multimer $s$ was derived to be:

$$s = \frac{2\pi\Delta r}{d_\beta}\left(\frac{\cos\beta_i \cos\beta_o}{n_o \cos\beta_i - n_i \cos\beta_o}\right) \qquad (1)$$

where $n_i$ is the number of strands per protomer in the inner β-barrel, $n_o$ is the number of strands per protomer in the outer β-barrel and $\Delta r$ corresponds to the difference between the inner β-barrel radius and the outer β-barrel radius, and is given by the intermolecular interactions of the β-hairpins; $d_\beta$ is inter-strand distance in β-sheets. $n_o$ is 3, $n_i$ is 2, $\Delta r$ is 9.4 Å and $d_\beta$ is 5 Å. On the basis of the currently solved β-barrels and idealized β-barrel structures, chain tilt angles can only span a limited range between 35° and 45° (ref. 42).

**Data availability.** The cryo-EM maps and the corresponding atomic models have been deposited to the Worldwide Protein Data Bank (http://www.wwpdb.org) with the following accession codes: EMD-8185 and 5JZH (Y221G mutant); EMD-8187 and 5JZT (K246C/E258C mutant); and EMD-8188 and 5JZW (wild-type protein). The additional data that support the findings of this study are available from the corresponding author upon request.

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

## Acknowledgements

We acknowledge the use of Netherlands Centre for Electron Nanoscopy (NeCEN) electron microscopes (Leiden University) funded by the Netherlands Organization for Scientific Research (NOW) and the European Regional Development Fund of the European Commission. We thank Nigel Unwin for critical reading of the manuscript. We thank Sylvia Ho for the help with protein purification and sample preparation. This project was supported by a Swiss National Science Foundation grant (#139098 to BZ). N.C. is supported by a fellowship of the Conselho Nacional de Desenvolvimento Científico e Tecnológico (CNPq) of Brazil. Part of the imaging was performed on equipment supported by the Microscopy Imaging Center (MIC), University of Bern, Switzerland. Computing was performed on the University of Bern Linux Cluster (UBELIX).

## Author contributions

I.I. designed the project, purified the proteins, performed the experiments, collected the data, performed image processing and model building, analysed the data and wrote the paper. S.D.C. collected the data at NeCEN and contributed in writing the paper. N.C. performed model building, analysed the data and contributed in writing the paper. M.D.P. analysed the data and contributed in writing the paper. F.G.v.d.G provided all protein plasmids, analysed the data and wrote the paper. B.Z. designed and supervised the project, performed image processing, model building, analysed the data and wrote the paper.

## Additional information

**Competing financial interests:** The authors declare no competing financial interests.

