## [Peer Review File · Nature Communications]

REVIEWERS' COMMENTS:

Reviewer #1 (Remarks to the Author):

The authors have addressed all the previous comments from the Nature submission. There are a few points to be addressed, but otherwise this work has significantly expanded our understanding of the pore forming process by the aerolysin-like family of proteins.

Minor comments

1. The authors make the statement at the end of the introduction: "...describe the entire pore formation process by the aerolysin family members: from the monomer to the oligomer, to the zipper-like formation of the β -barrel, to the final piston-like puncturing of the lipid bilayer." This is perhaps a bit over the top in the interpretation of the results and should be stated in a more precise manner as to the important aspects of aerolysin pore formation revealed by this work and how it may apply to other systems.
2. In the section "Structure of the aerolysin prepore" the authors should break up a single long paragraph into logical sections.
3. In the discussion the use of " β -inner barrel seems clumsy and should be changed "inner β -barrel"
4. The authors may want to expand their discussion a bit and make a few more comparisons with mechanisms of other unrelated pore forming toxins.

Reviewer #2 (Remarks to the Author):

The revision has addressed all of my previous comments. No further comments.